# The SILKIE (Skin graftIng Low friKtIon Environment) study: a non-randomised proof-of-concept and feasibility study on the impact of low-friction nursing environment on skin grafting success rates in adult and paediatric burns

Linda Hollén,[1,2] Rosemary Greenwood,[3] Rebecca Kandiyali,[1,2] Jenny Ingram,[2] Chris Foy,[4] Susan George,[1] Sandra Mulligan,[1] Francesca Spickett-Jones,[1] Simon Booth,[5] Anthony Sack,[6] Alan Emond,[1,2] Ken Dunn,[7] Amber Young[1]

For numbered affiliations see end of article.

**Correspondence to**
Dr Amber Young;
amber.young1@nhs.net

## ABSTRACT

**Objectives** To evaluate the impact of low-friction (LF) bedding on graft loss in an acute burn care setting, and to examine the feasibility and costs of using LF bedding compared with standard care.

**Design** Proof of concept before and after study with feasibility of delivering the intervention.

**Setting** Three burns services within two UK hospital trusts.

**Participants** Inclusion criteria were patients older than 4 weeks, who received a skin graft after burn injury and were admitted overnight. The comparator cohort were eligible patients admitted in a 12-month period before the intervention.

**Intervention** Introduction of LF sheets and pillowcases during a 15-month period.

**Outcome measures** For proof of concept, the LF and comparator cohorts were compared in terms of number of regrafting operations (primary), percentage graft loss, hospital length of stay (LoS) and LoS cost (secondary). Feasibility outcomes were practicality and safety of using LF bedding.

**Results** 131 patients were eligible for the LF cohort and 90 patients for the comparator cohort. Although the primary outcome of the proportion needing regrafting was halved in the LF cohort, the confidence interval (CI) crossed 1 (OR (95% CI): 0.56 (0.16 to 1.88)). Partial graft loss (any loss) was significantly reduced in the LF cohort (OR (95% CI): 0.27 (0.14, 0.51)). Inpatient LoS was no different between the two cohorts (difference in median days (95% CI): 0 (−2 to 1)), and the estimated difference in LoS cost was £−1139 (−4829 to 2551). Practical issues were easily resolved, and no safety incidents occurred while patients were nursed on LF bedding.

**Conclusions** LF bedding is safe to use in burned patients with skin grafts and we have shown proof of concept for the intervention. Further economic modelling is required to see if an appropriately powered randomised control trial would be worthwhile or if roll out across the National Health Service is justified.

## Strengths and limitations of this study

► A multicentre study confirming the feasibility, practicality and safety of using low-friction (LF) bedding in children and adults with skin grafts.
► Proof of concept that graft loss could be lower in an LF nursing environment has been demonstrated.
► A before-and-after design restricted the strength of evidence of the effectiveness of the intervention.
► Causality of using an LF environment for preventing graft loss could not be demonstrated.
► Lack of standardised reporting of partial graft loss limited conclusions of the benefit of the LF environment.

**Trial registration number** ISRCTN82599687.

## INTRODUCTION

Achieving wound closure early after burn injury results in improved survival, better cosmetic outcomes and shorter lengths of hospital stay.[1] To assist wound healing, surgical approaches with early tangential excision and wound closure using skin grafting are increasingly being applied to deep dermal or full-thickness burns.[2,3] Skin grafts are also used in partial-thickness burns failing to heal within 2 to 3 weeks.[4] There are roughly 1000 skin grafts undertaken in patients with burns annually in England and Wales, 75% undertaken in adults and 25% in children.[5] Integration of grafts to the wound requires vascular and lymphatic revascularisation and re-innervation.[6] Healing and 'take' depend on a well-vascularised clean recipient site, close apposition of the graft to the wound bed and

appropriate immobilisation of the graft to allow development of new vessels.[7] Suboptimal take (full or partial graft loss) will delay healing, increase hospital stay and may also require repeat surgery with increased pain and the potential for infection and increased scar formation with poor cosmetic outcomes.[8 9] These suboptimal outcomes cause distress to patients and impact negatively on the physical, social, emotional and economic aspects of their life.[10 11]

Loss of the graft typically occurs during the first few days and may be due to loss of contact with the wound bed due to haematoma, infection or shear.[7 8] It is thought that friction between the dressing and the environment can cause a shearing stress on newly transplanted skin cells and detachment from the wound bed. An even distribution of constant pressure on the graft is therefore likely to increase the chances of successful take.[12] However, maintaining stability and attachment of the graft to the wound bed can be difficult especially in mobile or semi-mobile patients including children.[13] Low-friction (LF) products have been shown to be clinically and cost-effective in the prevention of skin breakdown in other at-risk patients,[14] but not yet tested in patients with burns. LF products are currently in use in the National Health Service (NHS) in England for premature newborn babies, elderly patients and patients with neurological conditions to prevent pressure ulcers.[15]

We undertook this study using a before-and-after design to investigate whether an LF nursing environment, using bed sheets and pillow cases (LF (intervention) cohort), could improve skin grafting success rates in adult and paediatric patients with burns compared with standard care using normal bed sheets and pillow cases (comparator cohort). The main aims of this study were: (1) to assess primary and secondary outcome measures around graft loss to provide proof of concept for the use of this technology and (2) to examine the feasibility of our outcome measures and the practicality and safety of using an LF nursing environment in an acute burn care setting. We also assessed costs associated with the intervention (the purchase, maintenance and laundry of LF items; staff training in the use of LF items) and compared the two cohorts in terms of costs associated with inpatient length of stay (LoS).

## METHODS
### Study design
We used a non-randomised multicentre study to investigate the feasibility of our intervention. As part of this design, we undertook a before-and-after notes review comparing two cohorts, an LF cohort (intervention) and a comparator cohort (standard care), to provide proof of concept through the CI around effect size.

### Participants and data
For both the LF and comparator cohort, patients of more than 4 weeks of age with burns who received a skin graft and were admitted overnight were assessed for inclusion.

Patients were excluded if they were on a ventilator, needing inotropic support or required a vacuum-assisted closure dressing. Bedding was the only difference between standard care and the LF nursing environment in the form of LF sheets, cot sheets and pillowcases. Standardised operating procedures were written for the use of these materials both in the bed, in chairs and on equipment used to elevate the wounds. The sheets used were already on the market and available to the NHS.[15] They are made from a synthetic fabric similar to parachute silk, but washable at high temperatures and reusable. The sheets had a large central panel of the LF material with a 20.4-inch poly cotton border to the sides and 24.5-inch border top and bottom. The border was to reduce the likelihood of the sheets coming untucked as well as reducing the risk of sliding when patients got out of bed. The pillowcases had LF material on the top and polycotton on the bottom (to reduce the frequency of pillows sliding down the bed). The LF fabrics move freely and smoothly over patients' skin and the underlying surface. The sheets conformed to hospital infection control protocols. For the intervention, we expected a 3-month adjustment period; after which, the standardised operating procedures around the LF environment would be finalised. However, as no changes were made during this time, 15 months of data were used after the introduction of the LF environment. In participating units, the LF environment became standard care, allowing all eligible patients to be included in the intervention cohort. Clinicians could withdraw patients from the LF nursing if it was felt not to be in the best interest of their patient; however, the patient remained in the cohort to prevent biasing the comparison with the comparator cohort (intention-to-treat analysis). The comparator cohort data were collected for patients from a 12-month period prior to the introduction of LF bedding. Comparator patients were identified from theatre record books, the International Burn Injury Database (iBID)[5] and Trust information systems.

Data were collected in three burns services within two hospital trusts: one serving an adult population, one a paediatric population and the third both adult and paediatric patients. The same case report form (CRF) was used in both cohorts. Patients in the LF cohort were invited, by the site research nurse, to consent to complete questionnaires on quality of life and resource use and to take part in a telephone interview within 2 weeks of being discharged from the hospital to explore their views of the LF bedding.

### Outcomes
#### Primary proof-of-concept outcome
The primary outcome was skin graft failure rate, as assessed by the proportion of patients who underwent regrafting surgery.

#### Secondary proof-of-concept outcomes
We assessed two secondary outcomes: percentage graft loss and length of hospital stay. The method of recording

percentage graft loss varied between centres but was recorded as either the % graft failure, >90% healed or ≥95% healed. For the purposes of estimating % graft loss, we assumed that a graft that was 90% or 95% healed had less than 10% or 5% loss, respectively. LoS was recorded as the first inpatient LoS (total and per % body surface area (BSA)) and total burn service LoS including repeat admissions.

## Feasibility outcomes

We assessed two feasibility outcomes (recorded only in the LF cohort): practicality and safety. The practicality outcome was assessed through staff focus groups or telephone interviews, where any difficulties with bedding use on the wards, laundry problems and issues with absorbency and staining were discussed. In terms of safety outcomes, we collected data on falls or slips and wound infection. Formal risk assessments were carried out as required by each site for manual handling, tissue viability and infection control. The site leads received all safety incident reports for the designated clinical areas in 'real time'. Background data regarding the numbers of safety incidents were also known to the site leads. In addition to the standard sheeting material around the edge of the bed, further safety improvements included using neonatal cot sheets as shawls to cradle small children instead of full-sized sheets to reduce tripping hazards, and the provision of additional pillow cases to facilitate use on equipment rather than full-sized sheets. We used two measures of infection: presence of surgical site infections and wound infection as assessed through wound swab results.

## Expected sample size

The overall recruitment figure to the LF cohort was dependent on the number of patients who presented with burn injuries requiring skin grafting and an overnight stay at the three study sites during the recruitment period. This was estimated before study start from the International Burn Injury Database (iBID).[5] A figure of 200 over the recruitment period was initially predicted based on data from the previous 5 years. After detailed discussion with the database chair, however, it became apparent that the data provided from the database were for individual graft events, not patients, that is, 24% of grafts were multiple procedures in single patients. Expected recruitment numbers were therefore revised to 75% of 200, that is, 150 patients.

## Missing data/data quality

All data were anonymised and entered at each site onto a REDCap (Research Electronic Data Capture) database.[16] Initially, all sites used double data entry for input to REDCap and data were checked using the REDCap data comparison facility. Where there were discrepancies, the CRF was checked and the REDCap record amended to contain only correct data. A set of rules was agreed by the management team and applied to subsequent data entry which reduced the error rate and

all sites moved to single data entry. Missing data and error checking was continued throughout the study by checking all REDCap records against the CRF for every participant.

## Statistical analyses

Analyses were by intention to treat and performed in Stata V.14 (Stata Statistical Software: Release 14, StataCorp, College Station, Texas, USA, 2015). We compared the two cohorts at admission for several demographic and patient characteristic variables: centre admitted to, ethnicity, age, total BSA, gender, burn type, body location burnt, comorbidity present, American Society of Anesthesiology grade (physical status classification),[17] body weight, perioperative antibiotics given and time to first grafting procedure. Summary statistics used were proportions for categorical variables, and medians and IQR or means and SD for continuous variables. We used two-sample proportion tests to compare the proportion of all categorical demographic and patient characteristic and outcome variables in the two cohorts. OR and 95% CI are reported as a measure of effect size for differences in primary and secondary graft loss outcomes. For LoS outcomes, differences in medians and means with 95% CI are reported.

## Economic analyses

The economic work sought to provide information (mean costs and CI, where appropriate) on the acute healthcare resource use and costs associated with the new technology, with resources being valued using hospital price information, and national sources of unit costs for staff time and bed days.[18 19] Resource use data collected focused on: (1) the additional costs associated with the intervention (ie, the purchase, maintenance and laundry of LF items; staff training in the use of LF items); data collection on this element was limited to the LF cohort alone and (2) before-and-after comparison of inpatient LoS. In our analysis, based on reference costs,[19] we multiplied the cost of an 'inlier' bed day by LoS to estimate inpatient care costs at the level of the patient. To account for the skew in our cost data, we used non-parametric bootstrap methods to construct the CI around the incremental inpatient bed day cost.[20]

## Patient and public involvement

Patients and the public have been actively involved at all stages of this study. Early work was undertaken before study start to understand the views of patients with burns undergoing skin grafts. Parent and staff views on the sheet material were also sought ahead of the study at Bristol Royal Hospital for Children and further views were asked during a public engagement event midway through the study. Eleven children (age range 11–17 years) were involved in developing age-appropriate patient consent and information literature through an established young people's advisory group. Patient and public views on further

study patient documentation such as adult information sheet, parent information sheet and the consent forms were obtained from individuals who contacted the Burns Research Centre in Bristol via the 'People in Research' National Institute for Health Research website. Four members of the public also read and edited the adult patient documentation, so that the purpose of the project was clearer. Feedback on research design and outcome measures obtained was incorporated into the final study documents. The Ambassador for the Scar Free Foundation and burns survivor, read and edited the study protocol and advised regarding patient and public input. She was also involved in key decision-making through her membership and attendance in person at the steering group meetings. Patients were not actively involved in the recruitment to and conduct of the study. Following publication of results, we will disseminate to burn patient groups and the public through plain English-written documents.

## RESULTS

### Participant flow, recruitment and compliance

Recruitment to the LF cohort started on 6 October 2015 and was completed on 31 December 2016—a 65-week period. Of the 334 patients screened for eligibility in the LF cohort, 131 (39%) were eligible for LF nursing. For the comparator cohort, 90 eligible patients were identified across all three sites (CONSORT flow chart; figure 1).

More details on reasons for non-eligibility are provided in online supplementary appendix 1.

Of the LF cohort patients, 107/131 (82%) were nursed on the sheets for all (n=74) or part (n=33) of their inpatient stay. All patients using sheets for only part of their stay were inpatients for 6 days or longer and many returned to normal nursing after an initial period on the sheets. The reasons for 24 patients not being nursed on sheets included: surgeon preference (7/24: 29%) (this included, eg, patients with unstable fracture required to stay still, complicated burn location or concurrent illness), unplanned overnight stays as patients booked as day cases and not fit to be discharged (5/24: 21%), exclusion following risk assessment (4/24: 17%), no staff availability (3/24: 13%), staff forgetting (1/24: 4%), patient nursed in intensive care unit (1/24: 4%) and no explanation provided (3/24: 12%).

### Proof of concept

#### Comparison of cohorts at admission

Only one demographic difference was found between the two cohorts at admission (table 1). A higher proportion of patients receiving antibiotics perioperatively for graft surgery was reported in the comparator cohort compared with the LF cohort.

Baseline demographic data were also analysed for the subset of nursed versus not nursed on LF sheets within the LF cohort. The only differences found between the two subsets were that patients in the LF cohort not nursed

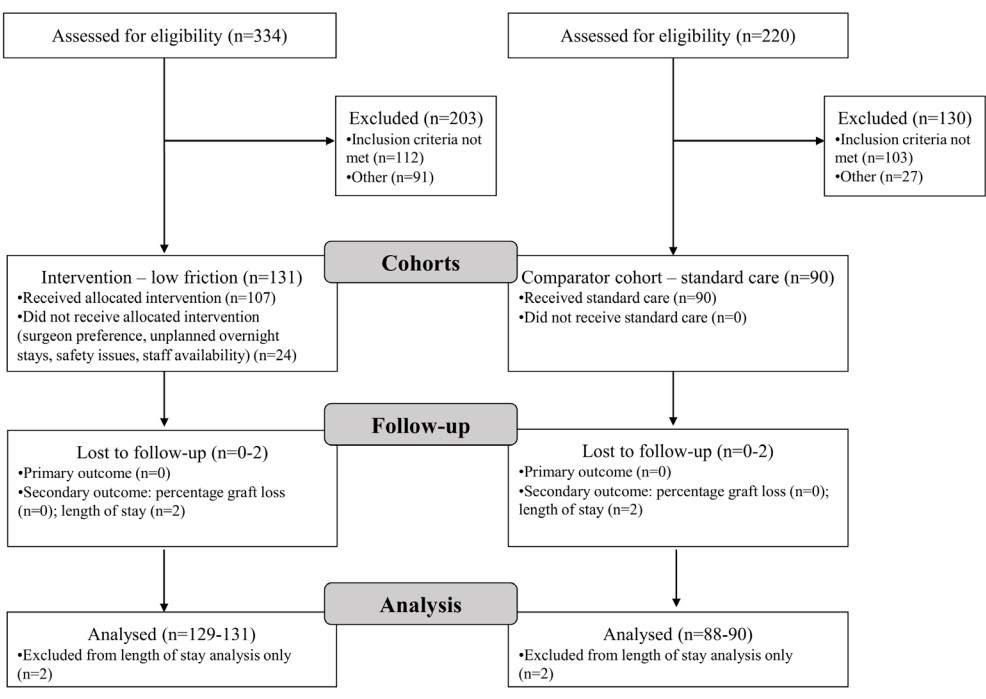

**Figure 1** Consort 2010 flow diagram for study.

**Table 1** Demographic differences between comparator (standard care) cohort and low-friction (LF) cohort

| | Comparator cohort | LF cohort |
|---|---|---|
| | (N=90) | (N=131) |
| Centre (proportion (%)) | | |
| A | 12/90 (13) | 27/131 (21) |
| B | 33/90 (37) | 51/131 (39) |
| C | 45/90 (50) | 53/131 (40) |
| Ethnicity (proportion white British (%)) | 77/90 (86) | 106/131 (81) |
| Age in years (median (IQR)) | 48 (23–62.5) n=88 | 41 (15–64) n=131 |
| TBSA (median (IQR)) | 3.3 (1.5–7.5) n=90 | 4 (1.5–10.0) n=131 |
| Gender (proportion male (%)) | 53/90 (59) | 78/131 (60) |
| Burn type (proportion scalds (%)) | 38/90 (42) | 37/131 (28) |
| Location (proportion (%)) | | |
| Head and neck | 24/90 (27) | 40/131 (31) |
| Anterior chest | 23/90 (26) | 34/131 (26) |
| Posterior chest or back | 13/90 (14) | 23/131 (18) |
| Upper limb | 29/90 (32) | 48/131 (37) |
| Hand or wrist | 21/90 (23) | 31/131 (24) |
| Abdomen | 8/90 (9) | 18/131 (14) |
| Buttocks | 10/90 (11) | 9/131 (7) |
| Perineum | 6/90 (7) | 2/131 (2) |
| Lower limb | 31/90 (34) | 63/131 (48) |
| Foot or ankle | 16/90 (17) | 26/131 (20) |
| Significant comorbidity (proportion yes (%)) | 32/89 (36) | 36/131 (27) |
| ASA grade (proportion normal healthy patient (%)) | 45/89 (51) | 59/131 (45) |
| Weight of patient (median (IQR)) | 69.5 (58.7–81.5) n=90 | 71.8 (55.3–86.0) n=131 |
| **Perioperative antibiotics (proportion yes (%))** | **73/90 (81)** | **85/131 (65)** |
| Time to first graft in days (median (IQR)) | 1 (1–5) n=90 | 2 (0–4) n=131 |

Text highlighted in bold shows 95% CI not crossing 1 in two-sample proportion tests.
ASA, American Society of Anesthesiology; TBSA, total body surface area.

on sheets had fewer posterior chest or back burns and a smaller proportion had antibiotics perioperatively for the first graft (table 2).

## Primary outcome
Data for the primary outcome of graft loss requiring regrafting was 100% complete in both the LF and comparator cohort. Graft loss requiring regrafting was 6.7% (6/90) with standard care and 3.8% (5/131) with LF nursing. The OR for this difference was 0.56 but 95% CI crossed 1 (table 3).

## Secondary outcomes
### Percentage graft loss
Data for the secondary outcome of percentage graft loss was 100% complete in both cohorts. A larger proportion of patients in the comparator cohort (40%) reported any graft loss compared with the LF cohort (15%) with an OR of 0.27 (95% CI 0.14 to 0.51) (table 3). Similar patterns were found when graft loss was measured as >5% or >10% reported loss (table 3).

### Length of stay
Two individuals in each cohort had missing data on this outcome. There was no difference between cohorts in any of the hospital LoS measures (table 3).

## Economic analyses
Across the three hospital sites, the total purchase costs for the LF items were £14 222 and staff training costs were £720. There were no additional charges to launder LF items. This equates to a mean intervention study cost of approximately £114 based on the 131 patients nursed in the LF environment. The mean LoS cost (SD) associated with the comparator cohort was £9608 (12 803) and £10 747 (14 365) for the LF cohort, with a mean difference of £−1139 (−4829 to 2551).

## Feasibility outcomes
### Practicality
In semistructured qualitative interviews, staff reported that putting the sheets on the bed was difficult as they became untucked due to the lack of 'fitted' design. They also found

**Table 2** Demographic differences between those nursed and not nursed on low-friction (LF) sheets within the LF cohort

| | On sheets | Not on sheets |
|---|---|---|
| | (N=107) | (N=24) |
| Centre (proportion (%)) | | |
| A | 22/107 (21) | 5/24 (21) |
| B | 39/107 (37) | 12/24 (50) |
| C | 46/107 (43) | 7/24 (29) |
| Ethnicity (proportion white British (%)) | 90/107 (84) | 16/24 (67) |
| Age in years (median (IQR)) | 41 (15–64) n=107 | 42 (20–61) n=24 |
| TBSA (median (IQR)) | 4 (1.5–10.0) n=107 | 2.5 (1–7.0) n=24 |
| Gender (proportion male (%)) | 64/107 (60) | 14/24 (58) |
| Burn type (proportion scalds (%)) | 33/107 (31) | 4/24 (17) |
| Location (proportion (%)) | | |
| Head and neck | 34/107 (32) | 6/24 (25) |
| Anterior chest | 29/107 (27) | 5/24 (21) |
| **Posterior chest or back** | **23/107 (22)** | **0/24 (0)** |
| Upper limb | 39/107 (36) | 9/24 (38) |
| Hand or wrist | 23/107 (22) | 8/24 (33) |
| Abdomen | 15/107 (14) | 3/24 (13) |
| Buttocks | 6/107 (6) | 3/24 (13) |
| Perineum | 2/107 (2) | 0/24 (0) |
| Lower limb | 52/107 (49) | 11/24 (46) |
| Foot or ankle | 22/107 (21) | 4/24 (17) |
| Significant comorbidity (proportion yes (%)) | 25/107 (23) | 11/24 (46) |
| ASA grade (proportion normal healthy patient (%)) | 50/107 (47) | 9/24 (38) |
| Weight of patient (median (IQR)) | 71.8 (54.2–86.0) n=107 | 71.5 (60.0–85.9) n=24 |
| **Perioperative antibiotics (proportion yes (%))** | **75/107 (70)** | **10/24 (42)** |
| Time to first graft in days (median (IQR)) | 1 (0–4) n=107 | 2 (1–5)n=24 |

Text highlighted in bold shows 95% CI not crossing 1 in two-sample proportion tests.
ASA, American Society of Anesthesiology; TBSA, total body surface area.

that the LF sheets were more difficult to keep clean and dry as the sheets were less absorbent and there was increased pooling due to the nature of burn wounds. However, the slipperiness of the sheets made it easier to move patients in bed. There were also some laundry issues due to the staining of the LF sheets, and sheet snagging. One site had initial issues with considerable laundry sheet loss, but all these issues were resolved and did not hamper the intervention. More detailed findings from staff and patient interviews about the sheets is reported elsewhere.[21]

### Safety

No tissue viability incidents were reported for any patient being nursed on the LF bedding and no surgical site infections were reported. There was also no difference between cohorts in the proportion of patients reporting a positive wound swab as a measure of infection (comparator: 71/87 (81.6%); LF: 103/131 (78.6%)). Each site reported one patient fall in the LF cohort; two were not using the LF bedding at the time and the third fell while suffering a myocardial infarction unrelated to the LF sheeting.

### DISCUSSION

In this study, we have provided proof of concept for the use of an LF environment in patients undergoing skin grafts for a burn injury. A consistent difference between the two cohorts before and after the introduction of LF bedding was seen across different ways of assessing partial graft loss. One in four patients with standard care sheets had at least 5% graft loss whereas only one in eight did on the LF sheets. Although the odds of needing a regrafting operation was halved in the LF cohort, the CI was wide and crossed 1 due to the low prevalence of regrafting operations in our study. A much larger study would be needed to provide the same effect size with narrow CI if the primary outcome of regrafting were to be used in a randomised controlled trial (RCT), necessitating the collection of data from over 2000 grafted patients. We showed no difference in LoS and associated costs between the two cohorts. Practical issues were mostly easy to resolve, and our results show that the LF bedding is safe for acutely burned patients undergoing skin grafts.

**Table 3** Proof-of-concept outcomes in the comparator cohort compared with the low-friction (LF) cohort

| | Comparator cohort (N=90) | LF cohort (N=131) | ES (95% CI)* | P values† |
|---|---|---|---|---|
| **Primary outcome** | | | | |
| Regrafted (proportion yes (%)) | 6/90 (6.7) | 5/131 (3.8) | 0.56 (0.16 to 1.88) | 0.33 |
| **Secondary outcomes** | | | | |
| Reported graft loss | | | | |
| **Proportion >0% (%)** | **36/90 (40)** | **20/131 (15)** | **0.27 (0.14 to 0.51)** | **<0.001** |
| **Proportion >5% (%)** | **22/90 (24)** | **15/131 (11)** | **0.40 (0.19 to 0.82)** | **0.01** |
| **Proportion >10% (%)** | **15/90 (17)** | **9/131 (7)** | **0.37 (0.15 to 0.88)** | **0.02** |
| Length of stay (LoS) | | | | |
| First inpatient LoS | | | | |
| Median (95% CI) | 6 (3–13) | 7 (3–16) | 0 (–2.00 to 1.00)‡ | 0.51 |
| Mean (95% CI)§ | 11.5 (15.4) | 12.2 (14.7) | –0.64 (–4.69 to 3.40) | |
| First inpatient LoS/% BSA | | | | |
| Median (95% CI) | 2 (0.9–4) | 2 (1.2–3.3) | 0 (–0.42 to 0.50)‡ | 0.87 |
| Total burn service LoS | | | | |
| Median (95% CI) | 33.5 (25.5–69.5) n=88 | 32 (18–51) n=129 | 5 (–2.00 to 12.00)‡ | 0.12 |

*For regraft and graft loss, OR and 95% CI are reported. For LoS, median and mean differences with 95% CI are reported. Text highlighted in bold shows 95% CI not crossing 1. Comparator cohort used as control cohort.
†P values stem from two-sample proportion tests or Mann-Whitney U-tests.
‡CI for generalised Hodges-Lehmann median differences.
§Used for cost analysis.
BSA, body surface area; ES, effect size.

Limitations with our study include the choice of a before-and-after design to test the intervention. Although this provides us with proof of concept data, only a full RCT could provide the best quality evidence regarding effectiveness of the LF bedding and the causality of the differences we observed. We recognise that the potential for bias is inherent in our chosen design and sought to mitigate this by comparing the demographic variables between the two cohorts. We did not find any patient or demographic characteristics which would suggest an alternative explanation for the differences in graft loss that we observed. Another limitation in this study was the lack of standardised reporting of partial graft loss across clinicians and sites. Following established local protocols, some sites were assessing graft loss, and some were assessing percentage healed, with inconsistency between whether it should be assessed as 90% or 95% healed. Consistency of assessment would allow accurate audits to take place allowing outcome success to be compared across sites and within sites which could have a major impact on clinical practice. However, despite this inconsistency, the effect size was consistent across the different ways of reporting.

As this type of LF bedding has not previously been used in patients with burns, comparing our results to existing literature is difficult. However, one study did assess the effectiveness of LF garments, made by the same material and company, in reducing skin breakdown in patients with pressure ulcers.[14] This study showed that there was a significant reduction in the proportion of patients who developed pressure ulcers following use of the LF garments. In addition, fewer patients admitted already with ulcers deteriorated when using the LF garments. This study also suggested that the savings associated with preventing skin breakdown outweighed the cost of the products used.[14]

### Implications and future research
Skin graft loss after surgery impacts on patients' outcomes and health service costs. Increased scarring due to the delay in healing will affect patients in terms of cosmetic, psychological and functional outcomes.[4] The magnitude of the effect will depend on the size of the graft loss relative to the original graft. Increased healthcare costs will be related to a need for reoperation, increased need for dressings and outpatient management and potential for infection and scar management. Formally assessing clinical effectiveness through an RCT could be problematic. Given the true low rate of regrafting, the trial would need to be prohibitively large. If the primary outcome was altered to be percentage graft loss, associated costs such as treating increased scarring would be harder to quantify than regrafting costs (reoperation, increased LoS). An alternative might be to consider the data provided by this study as sufficient proof of concept to begin a roll-out of LF bedding to burns services for use in grafted patients, and to conduct surveillance of this using routinely collected data. Although introducing such a new method into daily nursing may increase workload initially, it is unlikely to be a significant cost driver. At the time of our study, fitted sheets to fit the hospital beds were not available, so we had to use flat sheets which were more time-consuming

for nursing staff. If the sheets were to be adopted and a fitted version found to be acceptable, it is unlikely that additional nursing time would be required.

## CONCLUSION

This study has demonstrated that LF bedding is safe to use with burned patients undergoing skin grafting in acute care settings. Using the LF bedding did not prolong any patient's LoS and any practical issues arising in relation to sheet usage were relatively easy to resolve. The study showed good proof of concept for LF nursing, suggesting partial graft loss could be reduced if an LF approach was to become standard. The value for the NHS of mounting a full RCT remains in doubt, and economic modelling using value of information methods is currently underway before any further decisions are made.

**Author affiliations**

[1]The Scar Free Foundation Centre for Children's Burn Research, Bristol Royal Hospital for Children, University Hospitals Bristol NHS Foundation Trust, Bristol, UK
[2]Centre for Child and Adolescent Health, Bristol Medical School, University of Bristol, Bristol, UK
[3]Department of Research and Innovation, University Hospitals Bristol NHS Foundation Trust, Bristol, UK
[4]Gloucestershire Research Support Service, Gloucestershire Hospitals NHS Foundation Trust, Gloucester, UK
[5]McIndoe Burn Centre, Queen Victoria Hospital NHS Foundation Trust, East Grinstead, UK
[6]Adult Burns Unit, Southmead Hospital, North Bristol NHS Trust, Bristol, UK
[7]Department of Burns and Plastic surgery, The Manchester University NHS Foundation Trust, Manchester, UK

**Acknowledgements** We would like to thank everyone who has helped with this study, especially the patients, children and their parents for their time, effort and enthusiasm, the Research and Innovation, Infection Control and Tissue Viability teams at University Hospitals Bristol, North Bristol and Queen Victoria Hospital NHS Trusts, the burn surgeons' multidisciplinary teams and research nurses at all sites and the project steering committee. We thank Pam Warren, our patient representative for her invaluable help and active involvement with this project.

**Contributors** LH performed the analysis of data, contributed to development of methods, interpretation of results, and to the drafting and writing of the manuscript. RG led on development of methods, drafted the research protocol, interpretation of results and to critical reviewing of the manuscript. RK performed the costing analysis, contributed to development of methods, interpretation of results and to critical reviewing of the manuscript. JI contributed to development of qualitative methods, interpretation of results and critical reviewing of the manuscript. CF contributed advice on the costing analysis, contributed to development of methods, interpretation of results and to critical reviewing of the manuscript. SG, SM, FSJ, SB and AS collaborated to develop and lead the intervention, contributed to acquisition of data, interpretation of results and to critical reviewing of the manuscript. AE contributed to development of methods, interpretation of results, consensus meetings and to critical reviewing of the manuscript. KD contributed to the initial planning of the study, provided data from the International Burn Injury Database and critical reviewing of the manuscript. AY conceived and led the study and initiated development of the intervention, contributed to development of methods, interpretation of results and to critical reviewing of the manuscript. All authors contributed to and approved the final version and agree to be accountable for all aspects of the work.

**Funding** This work was supported by the National Institute for Health Research (NIHR) under its Research for Patient Benefit (RfPB) Programme (Grant Reference No. PB-PG-0214-33003).

**Disclaimer** The views expressed are those of the authors and not necessarily those of the NHS, the NIHR or the Department of Health. The Centre for Children's Burns Research is part of the Burns Collective, a Scar Free Foundation initiative with additional funding from the Vocational Training Charitable Trust VTCT and Health and Care Research Wales. The views expressed are those of the authors, and not necessarily those of the Scar Free Foundation or other funding bodies.

**Competing interests** None declared.

**Patient consent** Not required.

**Ethics approval** This study had the ethical approval from The National Institute for Social Care and Health Research Ethics Service Committee Wales REC 4 (REC reference: 15/WA/0156).

**Provenance and peer review** Not commissioned; externally peer reviewed.

**Data sharing statement** No additional data available.

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
