## [Reviewer comments · BMJ Open]

ARTICLE DETAILS

TITLE (PROVISIONAL)	The SILKIE (Skin grafting Low friction Environment) study: a non-randomised proof-of-concept and feasibility study on the impact of low friction nursing environment on skin grafting success rates in adult and paediatric burns.
AUTHORS	Hollen, Linda; Greenwood, Rosemary; Kandiyali, Rebecca; Ingram, Jenny; Foy, Chris; George, Susan; Mulligan, Sandra; Spickett-Jones, Francesca; Booth, Simon; Sack, Anthony; Emond, Alan; Young, Amber

VERSION 1 – REVIEW

REVIEWER	Christian Smolle, MD Division of Plastic, Aesthetic and Reconstructive Surgery, Department of Surgery, Medical University of Graz, Austria
REVIEW RETURNED	17-Mar-2018

GENERAL COMMENTS	The paper is very well written and states the research objective and results clearly. For the sake of better understanding, I missed only two aspects: First of all, I would recommend to attach p-values to all significant results, since they attract the readers attention and makes the results easier to understand, especially for those who are not familiar with the statistical tests used. Secondly, in my mind, a short paragraph explaining the forms of application of low friction sheets in the Methods section would be good. I.e.: are all components of bed linen covered with low friction sheets (pillow, blanket, sheet) or only parts of it. To sum up, after correction of those minor aspects the paper the paper should be accepted for publication.
--

REVIEWER	Moustafa Elmasry Department of Hand and Plastic surgery and Burn center, Linköping University hospital, Linköping, Sweden.
REVIEW RETURNED	19-Mar-2018

GENERAL COMMENTS	Thank you for a good and well planned study. I read the study and I have very few comments: 1-The study design is the main problem in this manuscript as addressed by the authors themselves; Before and after design is not the best method to assess such a new intervention and getting conclusions or recommendations based on this design is difficult. However, the authors were careful in their discussion and conclusion
--

	and this is to be taken in consideration. Actually, there is not much to do now when the study is ended, nevertheless, a bigger randomized control design is needed to verify this conclusions. 2- The costs calculation in burn care generally is difficult, different methods and scores are developed to measure this variable, however, none of them is satisfactory, I think when introducing such a new method in the daily nursing care this will account for some extra workload that improves with growing experience of the care providers , some estimation of the workload if possible would be beneficial in this context.
--	---

VERSION 1 – AUTHOR RESPONSE

Reviewer comments	Author responses
Reviewer: 1 The paper is very well written and states the research objective and results clearly. For the sake of better understanding, I missed only two aspects:	Thank you very much.
First of all, I would recommend to attach p-values to all significant results, since they attract the readers attention and make the results easier to understand, especially for those who are not familiar with the statistical tests used.	We have attached P-values to table 3 for the proof of concept outcomes.
Secondly, in my mind, a short paragraph explaining the forms of application of low friction sheets in the Methods section would be good. I.e.: are all components of bed linen covered with low friction sheets (pillow, blanket, sheet) or only parts of it.	This has now been added to methods under Participants and data section.
Reviewer: 2 Thank you for a good and well-planned study. I read the study and I have very few comments:	Thank you very much.

1-The study design is the main problem in this manuscript as addressed by the authors themselves; Before and after design is not the best method to assess such a new intervention and getting conclusions or recommendations based on this design is difficult. However, the authors were careful in their discussion and conclusion and this is to be taken in consideration. Actually, there is not much to do now when the study is ended, nevertheless, a bigger randomized control design is needed to verify this conclusions.	We fully agree and have tried to be careful in our conclusions.
2- The costs calculation in burn care generally is difficult, different methods and scores are developed to measure this variable, however, none of them is satisfactory, I think when introducing such a new method in the daily nursing care this will account for some extra workload that improves with growing experience of the care providers , some estimation of the workload if possible would be beneficial in this context.	We agree that costs for training with the new sheets will account for some extra workload initially but initial exploration suggests that training is not a significant cost-driver (under £1000 for 3 sites). Unfortunately, fitted sheets to fit the tip up beds were not available at the time of the study, so we had to use flat sheets that required more nursing time to put on. If the sheets were to be adopted and a fitted version found to be acceptable, it is unlikely that additional nursing time would be required. We have added a bit more on this in the discussion.

VERSION 2 – REVIEW

REVIEWER	Moustafa Elmasry Linköping university, Sweden
REVIEW RETURNED	02-Apr-2018

GENERAL COMMENTS	Thank you for responding to the comments. good luck in your further work.
---

REVIEWER	Christian Smolle, MD Medical University of Graz, Austria
REVIEW RETURNED	03-Apr-2018

GENERAL COMMENTS	With the changes made the manuscript is suitable for publication.
---